# Efficient Neural Networks for Tiny Machine Learning: A Comprehensive Review

## Abstract

The field of Tiny Machine Learning (TinyML) has gained significant attention due to its potential to enable intelligent applications on resource-constrained devices. This review provides an in-depth analysis of the advancements in efficient neural networks and the deployment of deep learning models on ultra-low-power microcontrollers (MCUs) for TinyML applications. It begins by introducing neural networks and discussing their architectures and resource requirements. It then explores MEMS-based applications on ultra-low-power MCUs, highlighting their potential for enabling TinyML on resource-constrained devices. The core of the review centres on efficient neural networks for TinyML. It covers techniques such as model compression, quantization, and low-rank factorization, which optimize neural network architectures for minimal resource utilization on MCUs. The paper then delves into the deployment of deep learning models on ultra-low-power MCUs, addressing challenges such as limited computational capabilities and memory resources. Techniques like model pruning, hardware acceleration, and algorithm-architecture co-design are discussed as strategies to enable efficient deployment. Lastly, the review provides an overview of current limitations in the field, including the trade-off between model complexity and resource constraints. Overall, this review paper presents a comprehensive analysis of efficient neural networks and deployment strategies for TinyML on ultra-low-power MCUs. It identifies future research directions for unlocking the full potential of TinyML applications on resource-constrained devices.

Keywords: Pruning; Compression; Deep learning; Efficient Neural Networks; Tiny Machine Learning

## 1 Introduction

**Artificial intelligence.** Over the last decade, *artificial intelligence* (AI) has revolutionized our daily experiences and technological advancements, empowering machines to perform tasks that traditionally require human-like intelligence, such as recognizing objects or speech or playing advanced games like Go.
*Machine learning* (ML) is the most prominent AI approach, which trains computers to learn patterns and representations from data without explicit programming.
*Deep learning* (DL) is an advanced subset of machine learning inspired by the organization of the brain, using artificial *neural networks* (NNs) to model and solve complex problems in a wide variety of fields, including language processing, protein generation, or automation.

**Sensors and microcontrollers.** Simultaneously, there has been an increase in the adoption and development of the *Internet of Things* (IoT), bringing new devices and applications into our daily

lives. *Micro-Electro-Mechanical Systems* (MEMS) and *Micro-Controller Units* (MCUs) are essential hardware components of IoT, which allows hardware devices to collect and process information (movement, voice, temperature, pressure...) directly at the source, in their local environment, excluding the need for additional resources or external communication. Local and autonomous data processing optimizes the flow of information but inherently poses power constraints. Some applications also require continuous data processing, which puts additional power constraints. MEMS and MCUs serve as the interface to sense information between the analog and the digital world. These devices are found in a wide range of applications, including mobiles, cars, wearables, environmental monitoring, and healthcare systems. Their consumer market scales to several billion in annual sales, so a slight deviation in power constraints can result in significant costs.

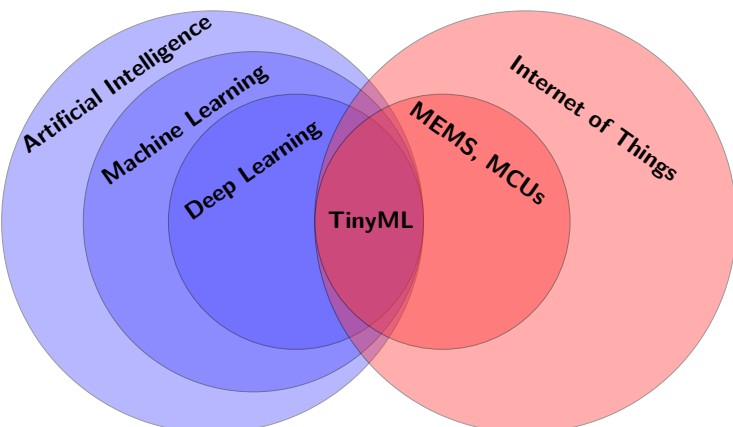

Figure 1: TinyML as the intersection between artificial intelligence and embedded systems.

**TinyML.** The convergence of machine learning and IoT has sparked significant interest in research and industry because it enables embedded hardware to process local data and interact with their environment in an automated and intelligent way. This intersection led to the emerging field of *TinyML*, a denomination first coined in 2019 by Han & Siebert (2022); see Figure 1. TinyML focuses on developing efficient neural network models and deployment techniques tailored for low-power, resource-constrained devices. In this context, the development of TinyML techniques aims to reduce the power consumption of neural networks *at inference time* while keeping good performance, which we refer to as *efficiency*. The power consumption essentially concerns both memory and computation resources. Memory usage is the first requirement to fulfill for ultra-low power devices, as the model cannot fit into the target hardware otherwise. Implicitly, we assume smaller models would be less demanding in computation for most cases, so our main objective is to target the lowest memory footprint.

Some examples of TinyML applications are detecting or counting events, gesture recognition, predictive maintenance, or keyword spotting, commonly found in home appliances, remote control devices, smartphones, smart watches, or augmented reality glasses.

However, the exponential growth of deep learning is closely linked to the development of powerful hardware, such as graphical processing units (GPUs), capable of supporting its large computation requirements. Therefore, deep learning has yet to reach the same growth and support on low-power devices, such as microcontrollers, to enable deep learning to run at the edge. Indeed, the power footprint of deep learning, as well as the vast landscape of embedded devices, pose new challenges but exciting opportunities that must be addressed by researchers and industrials.

**Objectives and outline.** The recent increase in research attention towards applying efficient deep learning techniques for ultra-low power devices has led to the emergence of several review articles, which can essentially be divided into two categories. The first category addresses more methodological

problems. Guo (2018) present both detailed theory and methods for quantization as well as a brief overview of neural networks, while Gholami et al. (2022) provide a more recent and thorough review of the methods. Gou et al. (2021) give an in-depth review of knowledge distillation methods, but not necessarily for tiny-sized neural networks. Hoefler et al. (2021) provide a thorough overview of pruning methods only for deep learning models. In addition, Alqahtani et al. (2021) also provide an overview of methods for efficient neural networks but misses an introduction to neural networks, to the hardware, and to the deployment process for tinyML. The second category of reviews focuses mainly on the applicative part. Han & Siebert (2022) provide a meta-review of tinyML papers and applications, while Schizas et al. (2022), Ray (2022) give a detailed overview of applications, hardware and deployment processes for tinyML, without introducing neural networks.

Unlike these works, we provide an accessible and comprehensive guide to TinyML, aimed at both deep learning practitioners and developers working with ultra-low power devices interested in integrating neural networks into microcontrollers.

The review begins with a general introduction to neural networks in Section 2, outlining their fundamental principles and architectures. It explores the evolution of neural networks and their applications in various domains, highlighting their computational requirements and the challenges they pose for resource-limited devices.

Then Section 3 presents a comprehensive overview of MEMS-based applications on ultra-low-power Micro-Controller Units (MCUs). It discusses the advancements in Micro-Electro-Mechanical Systems (MEMS) technology and its integration with MCUs, enabling the development of power-efficient sensing and actuation systems. The potential of MEMS-based applications in enabling TinyML on resource-constrained devices is emphasized.

The core of the review, Section 4, focuses on efficient neural networks for TinyML. This section examines various techniques and methodologies that aim to optimize neural network architectures and reduce their computational and memory requirements. It explores model compression, quantization, and low-rank factorization techniques, among others, showcasing their effectiveness in achieving high-performance inference on MCUs while maintaining minimal resource utilization.

Following the discussion on efficient neural networks, Section 5 delves into the deployment of deep learning models on ultra-low-power MCUs. It investigates the challenges associated with porting complex models onto MCUs with limited computational capabilities and memory resources. The section explores techniques such as model pruning, hardware acceleration, and co-design of algorithms and architectures, shedding light on strategies to enable efficient deployment of deep learning models for TinyML applications.

An overview of the current limitations in the field of TinyML is presented in Section 6. This section discusses the challenges faced by researchers and practitioners, including the trade-off between model complexity and resource constraints, the need for benchmark datasets and evaluation metrics specific to TinyML, and the exploration of novel hardware architectures optimized for TinyML workloads. Finally, Section 7 concludes and provides open challenges as well as insights into emerging trends and technologies that may impact the field of TinyML.

Overall, this review paper provides a comprehensive analysis of the advancements in efficient neural networks and deployment strategies for TinyML on ultra-low-power MCUs. It highlights the current state of the field and identifies future research directions necessary to unlock the full potential of TinyML applications on resource-constrained devices.

## 2 Neural networks

We introduce neural networks (Section 2.1), then we motivate how their theoretical properties (Section 2.2) and modern architectures (Section 2.3) are of interests in TinyML, and finally explain their implications for our work (Section 2.4).

### 2.1 Feedforward neural networks

The concept of artificial neural networks was introduced by McCulloch & Pitts (1943) as a mathematical model to simulate the human biological neural system but was limited in its ability to learn. This laid the foundation for the perceptron model, which was the first neural model capable of learning and classifying linearly separable data (Rosenblatt, 1958, Sakib et al., 2018). In turn, the backpropagation (Rumelhart et al., 1986) and gradient descent algorithms (Baldi, 1995, Lecun et al., 1998) were developed to allow efficient training of multi-layer perceptron (MLP) that is capable of classifying non-linear inputs. The MLP is a type of feedforward neural network that consists of alternatively stacking multiple layers $L$ of neurons and non-linear functions $\phi$ (Rumelhart et al., 1986, Huang, 2009) as represented in Figure 2. These layers include an input layer, one or more hidden layers, and an output layer. Stochastic gradient descent (SGD) and backpropagation algorithms, and progress in hardware computation have enabled the revolution in the field of neural networks, leading to the modern era of deep learning algorithms (LeCun et al., 2015), for example capable of achieving state-of-the-art performance on ImageNet (Krizhevsky et al., 2012).

Formally, a neural network can be defined as a function $f$ and a directed, weighted graph composed of nodes (neurons) and edges (connections between neurons) with associated weight parameters $W$, bias $B$, where inputs $x$ are propagated forward in the graph to produce an output $y$. The objective of the neural network $f$ defined as

$$
\begin{aligned}
y &= f(x) = h^{(L)} \\
h^{(l)} &= \phi^{(l)}\big(W^{(l)}h^{(l-1)} + B^{(l)}\big) \quad \text{for } l = 1, \dots, L \\
h^{(0)} &= x,
\end{aligned}
\tag{1}
$$

is to approximate some function $f^*$ mapping an input vector $x$ to an output vector $y$ by learning weights matrix $W$ (Goodfellow et al., 2016).

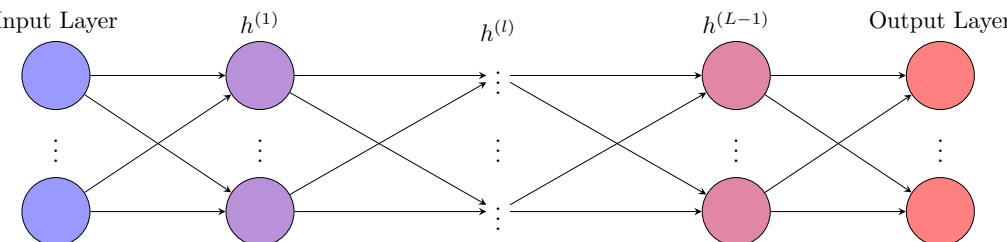

Figure 2: Feedforward neural network.

Neural networks have interesting theoretical and practical properties, as we will see in the next sections.

### 2.2 Properties

Neural networks possess powerful theoretical properties that stand out from standard machine learning approaches, making them of great interest for a wide range of applications.

**Expressiveness and generalization.** Neural networks are universal approximators: Cybenko (1989), Hornik et al. (1989) have theorized that a sufficiently wide hidden layer is able to approximate any continuous function on a compact set to an arbitrary level of precision. More recent work by Lin & Jegelka (2018) has extended the universal approximation theorem to residual neural networks (ResNets, He et al., 2015), proving that a sufficiently deep neural network with one-neuron hidden layers with residual connections has enough expressive power to approximate any continuous function.

Another crucial property in statistical learning theory in generalization (Vapnik, 2013, Vidyasagar, 2013), in which neural network models have shown that it is possible to generalize to new data with fewer examples than parameters with very large models (Li & Liang, 2018, Kawaguchi & Huang, 2019), and are even capable of labelling random data (Zhang et al., 2021a). This overparameterization results in a highly-dimensional non-convex space and redundancy, but results in higher quality and quantity of local minima (Choromańska et al., 2014). This implies that the optimization function has a higher chance of not getting stuck in a bad local minimum compared to small-size networks.

Over the recent years, research has revealed that enlarging the model size beyond the quantity of training examples can lead to a peculiar trend in test error: it may initially peak at a certain point of model complexity, then unexpectedly begin to decline again. This intriguing behavior has been termed *double-descent* by Belkin et al. (2019), who demonstrated its presence across various machine learning models, including a two-layer neural network. Further investigation into this phenomenon has been conducted by Nakkiran et al. (2021), who extensively explored double-descent in deep neural network models. They found that this trend can manifest when altering the model's width or the number of optimization iterations. Additionally, they observed instances of the double-descent phenomenon being influenced by dataset size, with larger datasets sometimes leading to inferior test performance.

Despite these findings, the underlying reasons behind the occurrence of double-descent in machine learning models and the specific inductive biases responsible for it remain incompletely understood. Nonetheless, it is crucial to consider this phenomenon when devising strategies aimed at enhancing generalization capabilities.

**Issues with expressiveness and generalization properties of TinyML.** These useful properties of are mainly verified for *large* neural networks. For instance, a two-layer neural network is ensured to be a universal approximator if its hidden layer is wide enough. On the generalization side, experimental and theoretical findings tend to show that larger models generalize better. However, in the TinyML setup, the neural networks are far from these large, well-studied models. Therefore, obtaining good training and generalization results in TinyML is very challenging.

After this brief overview of the theoretical properties of deep learning, we will now explore which modern deep learning architectures are commonly used in practice and why.

## 2.3   Modern deep learning architectures

Although in the modern deep learning era, the hardware progress can allow supporting the given high volume of computation and data, the design of the architecture is critical to the final performance and depends on the applications.

Developing and finding new neural network architectures is of great interest in research to surpass the state-of-the-art. Most of these state-of-the-art architectures are variations and combinations of the ones we present below. Table 1 provides a summary of standard architectures used in modern deep learning, and their strengths and weaknesses.

**Fully-connected layers.** Fully-connected (FC) layers, also known as dense layers were the first type of layers used in neural networks, specifically in MLP as presented in Section 2.1 and depicted

in Figure 2. Fully-connected connect each neuron of a layer to all the neurons in the next layer and process each input independently by applying a non-linear transformation. They are often used toward the end of the model to aggregate the higher-level features from the previous layers and make the final predictions. The simplest form of a fully-connected layer is a weighted sum, which makes them very general and not specialized to any particular application. Thus, they are building blocks of modern deep learning architectures. However, they are prone to overfitting, and may poorly perform on spatial or temporal data.

**Convolutional neural networks.** Convolutional neural networks (CNNs) are commonly used as feature extractors, showing their strength in processing spatial structures, such as images (Krizhevsky et al., 2012), videos (Simonyan & Zisserman, 2014), or signal processing (Alnaim & Abbod, 2019, Gong & Poellabauer, 2018). As the name suggests, they consist of applying convolutional operations using filters, also called kernels on the input in 1D, 2D or 3D, and are shared across the spatial dimensions. They are often stacked all together with max-pooling, to summarize a group of values by their maximum (Krizhevsky et al., 2012), batchnorm to normalize activations and facilitate training (Ioffe & Szegedy, 2015), and ReLU activation function. Compared to FC layers, this design allows CNNs to efficiently learn spatial hierarchical structures and detect local to global patterns, such as edges, shapes, and textures. In addition, the weight-sharing aspect reduces the number of parameters and makes them more robust to spatial translations and distortions. Some classic CNN architectures are AlexNet (Krizhevsky et al., 2012), VGGNet (Simonyan & Zisserman, 2015), or GoogLeNet (Szegedy et al., 2015), each using increasing network depths, thereby large model size.

Thus, in modern deep learning architectures, CNNs are often found in the early stages of the network serving as powerful feature extractors, but they have shown limitations in learning with sequential data structure or modeling long-range dependencies (Shorten & Khoshgoftaar, 2019, Liu et al., 2020).

**Recurrent neural network.** Recurrent neural networks (RNNs) are specialized layers for modeling sequential data (Rumelhart et al., 1986, Elman, 1990), such as signals (Graves & Jaitly, 2014, Alnaim & Abbod, 2019), speech (Zhang et al., 2018) or text (Bahdanau et al., 2015). Compared to CNNs, they are able to model longer temporal contexts by keeping a description of previous contexts because each output directly depends on previous inputs. This is of particular interest for sensor-based applications that inherently deal with sequential data.

The building block of an RNN can be defined as follows (Elman, 1990):

$$
\begin{aligned}
h_t &= \phi_h(W_h[h_{t-1}, x_t] + b_h), \\
y_t &= \phi_y(W_y h_t + b_y),
\end{aligned}
\tag{2}
$$

where $x_t$ is the input, $h_t$ is a shared internal state, serving as a *memory* at time $t$, $b_h$ and $b_y$ are bias terms, and $\phi_h, \phi_b$ are activation functions. However, they are difficult to train because of the effects of the vanishing or exploding gradient when the sequence is long (Bengio et al., 1994). Then long-short term memory (LSTM, Hochreiter & Schmidhuber, 1997, Gers et al., 1999; 2003) and gated-recurrent units (GRU, Chung et al., 2014) layers were designed to alleviate the limitations of the simple RNN.

They are based on two forms of memory updates:

- *Leak* memory updates, that are *progressive* updates of the current memory: $h_{t+1} = h_t + \phi(h_t, x_t)$,

- *Gate* memory updates, that are *context-dependent* updates of the memory: $h_{t+1} = \alpha h_t + (1 - \alpha)\phi(h_t, x_t)$,

where $\alpha$ can be a scalar or the output of a gated function $g(h_t, x_t) \in [0, 1]$ as in GRU or LSTM. Note that the "gated" mechanism is a specific form of the attention mechanism (Vaswani et al., 2017), allowing it to focus its attention on specific inputs depending on the context.

In particular, LSTM has three gates (input, forget, and output) and has two hidden temporal streams, one corresponding to the RNN stream of Equation equation 2, and another auxiliary stream used to compute $\alpha$, thus controlling the number of updates.

GRU is a simplified version of LSTM (update and forget) as well as one hidden temporal stream $h_t$, which has shown performance close to LSTM with a lower power footprint (Cahuantzi et al., 2021).

However, RNNs are limited in handling spatially structured data and processing sequences in parallel. This is because RNNs process input one time step at a time, Equation equation 2.

**Residual neural networks.** Residual neural networks (ResNets) were introduced in He et al. (2015). They provide each layer with direct feedback from distant previous layers to minimize the loss of gradient information during the backpropagation in deep networks. Although ResNets has shown state-of-the-art performance in computer vision (Khan et al., 2020), they are typically on the scale of millions of parameters (Menghani, 2023) and are more commonly applied on deep networks, which is not suitable for TinyML hardware.

**Transformers.** Transformers are attention-based models introduced in Vaswani et al. (2017) that surpass state-of-the-art performance on large-scale natural language processing tasks or computer vision tasks (Lin et al., 2022). They allow the models to focus their attention on each token of the input sequence (local) with respect to other tokens (global). This design addresses the limitations of CNNs and RNNs as stated previously because Transformers can process long-term dependencies and sequences in parallel. Although they have encountered great success and interest, they require a large amount of data, and a power footprint for both training and inference, even more than ResNets, which makes them bad candidates for TinyML.

Table 1: Summary of standard architectures used in modern deep learning.

| Layer & Definition | Strength | Weakness |
|---|---|---|
| *FC*: Connects all neurons in-between layers | High-level aggregations | Overfitting, not specialized |
| *CNN*: Conv. operations with shared parameters | Local and global spatial patterns | Struggles with sequences |
| *RNN*: Processes sequences with a hidden state | Temporal dependencies | Struggles with spatial patterns |
| *ResNets*: Deep nets with residual connections | Eases training deep networks | Large model size, expensive |
| *Transformers*: Self-attention for input relationships | Long-term local and global patterns | Large training data and power footprint |

**Activation functions.** Activation functions in deep learning introduce non-linearity to the model, enabling deep learning models to achieve higher levels of expressiveness and create more complex decision boundaries. This non-linearity is essential for processing real-world data, characterized by diverse and often non-linear features, effectively capturing intricate relationships within the data. Table 2 references standard activations used in modern deep learning.

**Regularization.** In Section 2.2, we have seen that neural networks possess interesting generalization properties. We will now explore popular regularization choices that help with generalization in practice.

As in standard machine learning, regularization can help neural networks to generalize better to unseen data, and make them less complex. Regularization techniques can either be of two forms, based on whether or not they directly alter the objective function:

- Explicit regularization:
  - $L_1$ penalizes the absolute values of the weights, encouraging sparsity, and thus simpler models,
  - $L_2$ penalizes the squared values of the weights, constraining their magnitude, and thus encourages smoother and simpler models.

Table 2: Reference table of standard activation functions.

| Name | Definition | Notes |
|------|-----------|-------|
| ReLU | $f(x) = \begin{cases} x, & \text{if } x \geq 0 \\ 0, & \text{otherwise} \end{cases}$ | Returns identity if positive, else 0 |
| Leaky ReLU | $f(x) = \begin{cases} x, & \text{if } x \geq 0 \\ \alpha x, & \text{otherwise} \end{cases}$ | Allows small negative values |
| PReLU | $f(x) = \begin{cases} x, & \text{if } x \geq 0 \\ \alpha_i x, & \text{otherwise} \end{cases}$ | Per-neuron learnable $\alpha_i$ values |
| Tanh | $f(x) = \frac{e^x - e^{-x}}{e^x + e^{-x}}$ | Returns value in range $(-1, 1)$ |
| Sigmoid | $f(x) = \frac{1}{1+e^{-x}}$ | Returns value in range $(0, 1)$ |
| Softmax | $f(x_i) = \frac{e^{x_i}}{\sum_{j=1}^{K} e^{x_j}}$ | Returns class probabilities |

- Implicit regularization:

  - Dropout (Srivastava et al., 2014a) as an average of probabilistic architectures where each dropout-realization results in a different sub-network (Gal & Ghahramani, 2016),
  - Batch normalization limits the range of values and adds noise to the activation, preventing the model from memorizing the training data too well (Ioffe & Szegedy, 2015, Bjorck et al., 2018),
  - Early-stopping prevents the model from becoming too specialized during training (Sjöberg & Ljung, 1992, Bishop, 1995),
  - Data augmentation increases the size and diversity of the training set, which helps the model learn more robust features (Shorten & Khoshgoftaar, 2019),
  - Random noise injected into the input (also a form of data augmentation) (Goodfellow et al., 2016),
  - Noise introduced by SGD optimization (Poggio et al., 2020a;b).

Most of these regularization methods add negligible computation costs and help with generalization performance.

In this section, we provided a brief overview of the layers used in modern deep learning and discussed which have the most potential for low-power hardware applications.

## 2.4 From large deep learning models to TinyML

In this section, we give an overview of the recent trends of deep learning model sizes, then we explicit the challenges of TinyML based on the neural network theory (Section 2.2) and practices (Section 2.3), and motivate our interest to apply them for TinyML.

**Trend in deep learning models.** Since the first AlexNet model was trained on a graphic processor unit (GPU) (Krizhevsky et al., 2012), we entered the modern era of deep learning where the limits of the state-of-the-art are regularly pushed on numerous complex tasks. Meanwhile, deep learning models are geared towards exponential increases in model size. As of 2023, the GPT4 model (OpenAI, 2023) is said to be even larger than the GPT3 model with 175 billion parameters ($\approx$ 800GB) (Brown et al., 2020), being about 2800 times larger than AlexNet size in just over 8 years.

Although model performance can benefit from overparameterization, large neural networks have been shown to have high redundancy (Han et al., 2016, Frankle & Carbin, 2018). Denil et al. (2013) estimated that in some cases only about 5% of the total parameters are critical to the final output decision. Thus, we can see that these models fail in terms of *algorithm efficiency*, where the objective is to achieve a task with minimal effort.

This raises questions on how to train more efficient models and also suggests the existence of smaller but viable models.

**Trend in efficient deep learning models.** A new wave of efficient deep learning models emerged, such as SqueezeNet (Iandola et al., 2016), MobileNet V1, V2, and V3 (Howard et al., 2017, Sandler et al., 2018, Howard et al., 2019), or EfficientNet (Tan & Le, 2019), ranging in one to five million parameters, entering the scale of the feasibility on mobile devices. These new models can achieve up to a 510-time model size reduction compared to AlexNet (Tan & Le, 2019) with equal performance. In general cases, model sizes are in the order of at least $10^6$.

**Trend in ultra-low-power deep learning models.** Although mobile-sized models show a great shift toward efficient deep learning architectures, they are still too large for deployment on microcontrollers (Liberis & Lane, 2020, Lin et al., 2020, Banbury et al., 2021b). Deep learning on microcontrollers (Unlu, 2020) is an alternative paradigm that is still at an earlier stage compared to mobile-size research, where the term TinyML has been first appearing in 2019 (Han & Siebert, 2022). However, there has been a success in the deployment of neural networks on MCUs on audio classification tasks (Zhang et al., 2018, Lin et al., 2020, Fedorov et al., 2020) by using efficient CNNs, RNNs, or NAS (Banbury et al., 2021b). In Lin et al. (2020), they succeeded in deploying a person detection model with less than 1MB memory. In general cases, model sizes must be in the order of less than $10^6$ and less than 1MB. These models reach a memory size of under 512 kB or even 256 kB, entering the scale of microcontroller hardware. The high resource limitations of MCUs present unique requirements and need the design of dedicated workflow and tools to enable end-to-end deep learning pipelines. Table 3 provides a summary of example model sizes for each platform we reviewed.

Table 3: Comparison of representative deep learning model sizes across cloud, mobile, and MCU platforms.

| Platform | Model | Parameters | Model size |
|----------|-------|------------|------------|
| Cloud | Inception-v3 | $> 10^7$ | $> 100$MB |
| Mobile | MobileNet-v3 | $10^6$ | $> 1$MB |
| MCU | MCUNet | $< 10^6$ | $< 1$MB |

**Motivations.** Neural networks are powerful algorithms that can operate with an end-to-end approach in terms of algorithm design: labelled data, automated feature extraction and modeling, and deployment, for a wide range of applications. This makes them a great class of algorithm candidates for MEMS-based applications relying on signal processing. Unfortunately, the expressiveness and generalization ability of neural networks are dependent on their size, which makes them inherently complex and "black box" functions that are analytically difficult to interpret and design. However, they are mostly composed of very primitive operations, see Equation equation 1: *multiplications and additions*, which are accessible to any microcontrollers. Concerning the non-linear activations, some are very straightforward, such as ReLU (Fukushima, 1975, Nair & Hinton, 2010) or LeakyReLU (Maas et al., 2013), while other activations like tanh or sigmoid pose more challenges due to their computational complexity.

Moreover, prior literature has shown that it is *possible*, albeit *challenging*, to design and deploy small enough neural networks on resource-constrained microcontrollers. Therefore, following the trend of

efficient deep learning models to reduce their inherent power footprint, we are interested in pushing the state-of-the-art of low-power footprint models to make them viable to microcontrollers, without degrading performance. Additionally, deep learning models in practice are commonly overparameterized (Denil et al., 2013), so the field of deep learning will benefit from more contributions to designing and deploying more efficient and accessible neural networks.

To summarize, we provided background on neural network theory and practices, their limitations and challenges, and why they are of great research interest for MEMS-based applications running in ultra-low-power settings.

Next, we explore the literature on specialized methods to design efficient deep learning models for TinyML in Section 4, but we must first provide the necessary background on embedded hardware, which we will reference throughout our work in Section 3.

## 3 MEMS-based applications on ultra-low-power microcontrollers

We provide a brief overview of MEMS and MCU hardware technology (Section 3.1) to understand the scope of applications (Section 3.2) and their intrinsic challenges for deep learning (Section 3.3).

### 3.1 Overview

**MEMS and MCUs.** MEMS are miniaturized (microscale dimensions) sensors and actuators omnipresent in a wide range of electronic devices, as they convert physical and analog information into digital inputs about their local environment (Lammel, 2015, Zhu et al., 2020), that can be processed by MCUs in real-time. Some examples of MEMS are accelerometers, microphones, or pressure sensors. Table 5 provides examples of different sensor types and their applications. Thus, they provide an interface to sense real-world information from hardware to software.

MCUs are miniaturized computers that are non-invasive ($\sim 1$ mm$^2$ silicon area), cheap ($\sim 1$\$), low-power ($\leq 0.5$ W), and are dedicated to performing one task for months or even years within a device (Banbury et al., 2021b, Garbay et al., 2022). MCUs are composed of connectors, input/output interface, on-chip storage (ROM), volatile memory (SRAM) for intermediate data, and a CPU with a frequency usually below the $10^3$ MHz range (Banbury et al., 2021b). With over 250 billion MCUs already in use, forecasts predict a volume of 38.2 billion in 2023 alone (Lin et al., 2020). In this context, we emphasize that even a small difference in the power footprint between low-power hardware targets can translate to several billions of dollars in savings for the consumer market. This is exemplified by the 2\$ difference observed between the low-end of MCUs in Table 4. Even between MCUs, there are several orders of magnitude in terms of low power (Table 4). For example, the Cortex-M4 only consumes 0.1W, yet it still represents a target that is 1500 times more power-hungry and 20 times more memory (SRAM) capacity compared to the Cortex-M0+. Additionally, it is three times more costly for consumers. Consequently, it is important to highlight the strong industrial incentive to target the low-cost and low-power consumer market as much as possible with tiny hardware targets. By focusing on the power scale between these targets, we can realize billions in cost savings and other benefits that low-power MCUs offer for the consumer market.

**Applicability.** Sensing data at the edge allows for offline operations, as opposed to using online cloud computing, always-on and real-time processing, no network latency, limited energy overhead, and inherent privacy. MCUs are ubiquitous in modern electronic devices, including cars, mobiles, TVs, and cameras. Their high volume in the consumer market and wide applicability reinforce the significance of research and industry efforts in TinyML applications.

In this work, we target the most *extreme low-end range* of MCUs, with less than 8kB of RAM and 10MHz processing speed for extreme low-power deep learning inference. Therefore, we aim to push

Table 4: Comparison of hardware for **Cloud**, **Mobile**, and **TinyML** platforms (Banbury et al., 2021b, Saha et al., 2022). The three architectures studied in Section 6 are highlighted in blue.

| Platform | Architecture | Memory | Storage | Frequency | Power | FLOPS | Price |
|---|---|---|---|---|---|---|---|
| **Cloud** | *GPU* | *HBM* | *SSD/Disk* | | | | |
| Nvidia V100S | NVIDIA Volta | 32GB | TB$\sim$ PB | 1.2$-$1.3GHz | 250W | $\sim$16.4G | 14500$ |
| **Mobile** | *CPU* | *DRAM* | *Flash* | | | | |
| Galaxy Note 20 | Kryo 585 | 8GB | 128GB | 1.8$-$3.1GHz | $\sim$8W | 1.2T | 550$ |
| **TinyML** | *MCU* | *SRAM* | *eFlash/ROM* | | | | |
| SAME70Q21B | Cortex-M7 | 384kB | 2048kB | 300MHz | 0.3W | $\sim$432M | 5$ |
| SAMG55J19 | Cortex-M4 | 160kB | 512kB | 120MHz | 0.1W | $\sim$180M | 3$ |
| Newport | Cortex-M0+ | 8kB | 16kB | 6.14MHz | 70$\mu$W | N/A | 1$ |
| Newport | eDMPv1 | 4kB | 16kB | 6.14MHz | 66$\mu$W | N/A | 1$ |

the hardware limit that is currently not considered in the state-of-the-art for embedded deep learning. In particular, we focus on the common ARM Cortex-M series microcontrollers (Yiu, 2019), and particularly the Cortex-M0+ and M4 (Table 4), or the eDMPv1 depending on the application.

## 3.2 Scope of applications

As previously stated, the ability to embed neural networks at the edge can already benefit a wide variety of applications and can potentially lead to completely new types of products (Kanjo, 2022).

Common applications are image detection, gesture recognition, such as human activity recognition (HAR), or keyword spotting. Note that these are all wireless applications, that must operate in real-time and are always-on. In this context, the device returns a decision at all times, so it is expected to provide a seamless user experience (e.g., not missing any user intention (false negatives) or over-triggering (false positives)). Their sensor types and target devices are specified in Table 5.

Table 5: Example of sensor applications and their target MCU devices.

| Sensor types | Applications | Target devices |
|---|---|---|
| Accelerometer, Gyroscope, Magnetometer | Human activity recognition, gesture recognition, motion detection, voice detection, predictive maintenance | Arm Cortex-M0+ |
| Pressure | Fingerprint detection | Arm Cortex-M0+ , Cortex-M4 |
| Microphone | Sound classification, keyword spotting | Arm Cortex-M4 , Cortex-M7 |

## 3.3 Challenges of ultra-low-power hardware

Compared to mobile devices, the all-on-chip design, as shown in Figure 3, allows the processing of data at the closest location to the source, resulting in lower communication latency and lower power consumption. Thus, this is ideal for real-time and low-power constraints. However, it also makes them inherently constrained because additional memory cannot be extended with an SD card for example.

Moreover, Table 4 highlights that the Cortex-M0+ and M4 are among the most resource-constrained devices, with the Cortex-M0+ lacking support for floating-point operations. Consequently, we restrict to fixed-point (in contrast to floating point) values (Figure 4) and arithmetic which approximates real-values and computations (Menard et al., 2006), to comply with the inherent hardware and energy

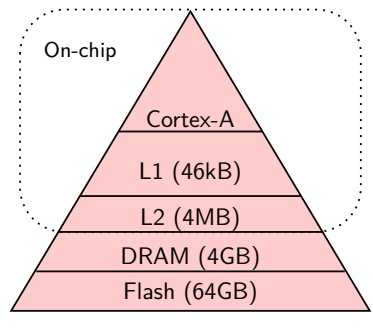 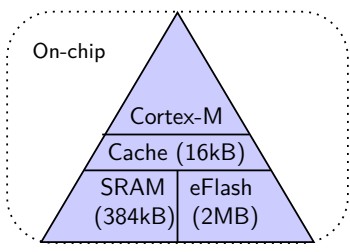

(a) Mobile processor          (b) Arm Cortex-M7 microcontroller

Figure 3: Illustration of memory hierarchies for (a) a mobile processor and (b) an Arm Cortex-M7 microcontroller (right). The microcontrollers process all computation and data transfer on-chip.

constraints of MCUs. Floating-point to fixed-point conversion requires a scaling factor of a power of two, which can be inferred as a simple bit shift and rounding as follows:

$$
\begin{aligned}
Q(F, n) &= \lfloor F * 2^n \rceil \\
F(Q, n) &= Q * 2^{-n}
\end{aligned}
\tag{3}
$$

where $Q$ and $F$ are the fixed point and floating point numbers, respectively, and $n$ is the number of bits. In practice, this means that we are limited to *integer-only operations*. Thus, only primitive operations like bit-manipulation, boolean operators, and basic additions or multiplications are supported in contrast to computationally intensive operations, such as explicit division or exponentiation. Additionally, the memory is typically the first bottleneck, so we seek lower-bit precision parameters than 32-bits, but this may increase the risk of overflow, or numerical precision loss and thus erroneous inference. From a hardware point-of-view, restricting to integer-only inference removes the need for a floating-point unit, which saves silicon area for each embedded chip, and thus billions of dollars of annual savings.

| 31 | 30 29 28 27 26 25 24 23 | 22 21 20 19 18 17 16 15 14 13 12 11 10 9 8 7 6 5 4 3 2 1 0 |
|---|---|---|
| sign (1-bit) | exponent (8-bits) | significand (23-bits) |

(a) Single precision floating-point 32-bit representation from IEEE754 (IEEE, 2019).

| 31 30 29 28 27 26 25 24 23 22 21 20 19 18 17 16 | 15 14 13 12 11 10 9 8 7 6 5 4 3 2 1 0 |
|---|---|
| integer part ($m$-bits) | fractional part ($n$-bits) |

(b) Fixed-point Q16.16 (Q$m.n$) on 32-bit representation.

Figure 4: Floating-point and fixed-point 32-bit representations. Floating-point (b) allows a dynamic range (minimal to maximal possible value) of roughly $[-10^{38}, 10^{38}]$, compared to fixed-point (a) $[-2^{m-1}, 2^{m-1} - 2^{-n}] \approx [-2^{15}, 2^{16}]$, which is approximately a $10^{33}$ smaller range (Novac et al., 2021). The smallest resolution (step between each consecutive representable value) of floating-point is $\approx 10^{-38}$ while it is $[2^{-n}] \approx 10^{-5}$ for $n = 16$ for fixed-point.

After a comprehensive review of the literature, the Cortex-M0+ and eDMPv1 appear to be one of the most resource-constrained platforms on which successful implementation of state-of-the-art deep learning has been reported (Zhang et al., 2018, Banbury et al., 2021b, Saha et al., 2022). Zhang et al.

(2018) deployed a 70kB keyword spotting application on an Arm Cortex M7, while Banbury et al. (2021b) deployed the same application on an Arm Cortex-M4 with a higher accuracy.

Furthermore, embedded hardware has a very heterogeneous ecosystem because specifications may differ from one manufacturer to another, and even between new series of the same brand, making it challenging to find common tools and approaches that are widely supported.

Therefore, the ultra-low-power hardware context presents a unique set of challenges due to their inherent resource limitations. Addressing these challenges poses high research and industry potential value and can lead to transformative advancements in real-time and low-power applications across numerous domains.

To summarize, in Section 2 and Section 3 we provided background on neural networks, and low-power sensors and motivated the challenges and objectives of our work. We will now examine the literature on methods (Section 4) and tools (Section 5) to design and deploy efficient neural networks for MEMS-based applications.

## 4 Efficient neural networks for TinyML

Building upon the concepts and motivations surrounding neural networks and embedded systems introduced in the previous sections, we now turn our attention to their intersection: TinyML.

This emerging field aims to combine the powerful benefits of neural networks with the cost-effectiveness of ultra-low-power devices with limited power, memory, and processing capabilities. Given the constraints of TinyML, developing efficient neural network architectures and algorithms is essential. In light of the growing efforts in this area, there is an increasing need for methods that can effectively scale to the most challenging embedded hardware, particularly in the context of MEMS-based applications.

In this section, we explore the methods available to train and design efficient neural networks for deployment on MCUs, enabling the deployment of intelligent applications on low-cost devices.

**Efficient RNNs.** Sensor applications mainly process time-related data continuously, so we are naturally interested in standard RNN layers, such as RNN (Rumelhart et al., 1986, Elman, 1990), GRU (Chung et al., 2014), LSTM (Hochreiter & Schmidhuber, 1997). Arık et al. (2017), Bhardwaj et al. (2022), Lu et al. (2022) have used convolutional recurrent neural networks (CRNNs) with a GRU or LSTM as the recurrent layer for keyword spotting or motion recognition applications for low-power and real-time inference, which matches our target applications and environment. The CRNN architecture offers strengths both in feature extraction, and time sequence processing, as well as compatible size for our target hardware (Bhardwaj et al., 2022).

In particular, Arık et al. (2017) empirically showed that GRU layers offer better size-performance tradeoff over LSTM in keyword spotting applications, which is our most demanding use case.

Moreover, there have been research efforts to find efficient alternatives to standard RNNs, such as minimal RNN (Chen, 2018), minimal gated unit (MGU) (Zhou et al., 2016), MGU1, MGU2, MGU3 (Heck & Salem, 2017). The MGUs differ from GRUs by reusing the gates, removing the bias term or the weight matrix completely, or a combination, detailed as follows:

$$
\begin{aligned}
\text{MGU1:} \quad & f_t = \phi\left(U_f h_{t-1} + b_f\right) \\
\text{MGU2:} \quad & f_t = \phi\left(U_f h_{t-1}\right) \\
\text{MGU3:} \quad & f_t = \phi\left(b_f\right),
\end{aligned}
\tag{4}
$$

where $f_t$ is the unique gate of the recurrent unit with weight parameters $U_f$, bias $b_f$, and $h_{t-1}$ the previous hidden state.

We notice that the MGU1, MGU2, and MGU3 variants do not directly gate the current input $x_t$, but instead, they indirectly gate the previous input $x_{t-1}$ by gating the previous state $h_{t-1}$, that has processed the previous input $x_{t-1}$.

Zhou et al. (2016), Heck & Salem (2017) suggest that these alternatives are competitive with GRU in terms of accuracy with a smaller parameter budget and thus should be more low-power friendly.

Next, we explore the methods that apply directly to models in order to reduce their power footprints.

**Model compression techniques.** Model compression is a set of methods aiming to address the growing power footprint and costs associated with the deployment of neural networks in terms of size and computation on resource-constrained devices (Neill, 2020, Hoefler et al., 2021), such as MCUs. In the following sections, we will provide an overview of the most commonly used techniques, which essentially encompass five methods: knowledge distillation, pruning, quantization, weight-sharing, and low-rank matrix decomposition (Neill, 2020).

### 4.1 Knowledge distillation

Knowledge distillation is a high-level approach to model compression, first explored in Buciluă et al. (2006) to reduce the model size by learning a small (student) model from an ensemble of models (teacher). Then Hinton et al. (2015) popularized knowledge distillation for neural networks where a small model (student) is trained from the supervision of a larger and overparameterized trained model (teacher) that has learned "dark knowledge". The idea is to leverage the latent knowledge the large teacher has captured and transfer it to the student during the training process. The loss encompasses both the original student loss (e.g., cross-entropy) and the difference between the teacher and student distribution, expressed as follows:

$$L_{\text{KD}}(x, y) = \alpha L_{\text{S}}(x, y) + (1 - \alpha) \text{D}_{\text{KL}} \left( \text{softmax} \left( \frac{T(x, y)}{\text{temp}} \right), \text{softmax} \left( \frac{S(x, y)}{\text{temp}} \right) \right), \quad (5)$$

where $L_S$ is the student loss function, $S(x, y)$ is the output of the student model, $T(x, y)$ is the output of the teacher model, $\text{D}_{\text{KL}}$ is the Kullback–Leibler (KL) divergence, $\alpha \in [0, 1]$ is a hyperparameter that controls the amount of distillation given by the teacher to the student, and temp is another hyperparameter that softens the probability distributions of the output models.

In practice, we must choose and train one teacher and one student architecture. Hinton et al. (2015) showed promising results across general computer vision tasks and sequential data. However, the disadvantages are that it requires empirical knowledge to find good teacher and student models, as well as additional computations to train the teacher and the forward pass of the teacher during the student's training. Although the design of the teacher would consist of training an overparameterized model, which works well in practice, the student should be the size of our target model. Moreover, we can bypass the additional forward pass of the teacher by storing its output along with the training set.

Therefore, the general framework design of knowledge distillation is flexible for our case and has proven promising performance in a wide range of applications.

### 4.2 Model pruning

While knowledge distillation involves training a new smaller model, pruning focuses on removing less important parts of a model. From a neuroscience perspective, the human brain has a pruning mechanism that removes redundant connections or irrelevant information from past experiences (Walsh, 2013, Neill, 2020). In the case of deep learning models, they are notoriously overparameterized (Section 2.2), which provides them with a large degree of freedom. In fact, it has been found that only a small fraction of the total parameters are critical (Denil et al., 2013). Model pruning is a very active research area at the intersection of promoting efficient deep learning and understanding neural network

training and generalization ability, where new methods emerge continuously (Alqahtani et al., 2021, Hoefler et al., 2021, Freire et al., 2023).

Han et al. (2016), Ullrich et al. (2016) made a major breakthrough for model compression in the modern deep learning era, where they combined pruning, quantization (Section 4.3) and Huffman encoding (Huffman, 2006) to reduce a CNN model by 49 times its size with less than 0.5% accuracy loss on the ImageNet dataset.

Seminal work by Frankle & Carbin (2018), Liu et al. (2018b) provided more theoretical understanding; the lottery ticket hypothesis (LTH) states that there exists a sparse subnetwork (winning ticket) that can be trained from scratch with the same initialized weights and reach the performance of the original network (10 times larger). In this view, a large model has a greater chance of containing a good subnetwork. They suggest that the network architecture itself is more critical than keeping the values of the weights in the original trained network. In practice, Frankle & Carbin (2018) requires iterative pruning trials of subnetworks to find the winning ticket, which is computationally expensive. Further work extended the LTH, showing that universal tickets could be reused across other applications (Burkholz et al., 2022, Fischer & Burkholz, 2022). In particular, Ramanujan et al. (2020) generalized the LTH to the strong lottery ticket hypothesis (SLTH) where the subnetwork performs well with the randomly initialized parameters and thus does not require retraining. Additionally, Burkholz et al. (2022) demonstrates that SLTH can also yield universal tickets across other applications. Consequently, the SLTH promises that training deep learning models could be replaced by efficient neural network pruning (Fischer & Burkholz, 2022). Alternatively, pruning can be seen as a form of neural architecture search (NAS) (Elsken et al., 2019), aiming to find Pareto-optimal architectures (Liu et al., 2018b). Moreover, it is also a form of regularization because it reduces the complexity of the model, similar to dropout, but the effect remains permanent.

There are essentially two types of pruning: unstructured and structured pruning, referring to how the pruning is performed in a weight matrix of a model, as illustrated in Figure 5.

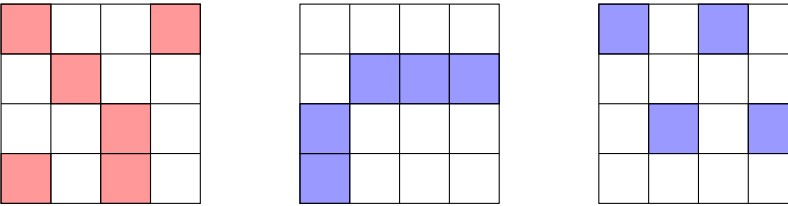

Figure 5: Unstructured pruning (left panel) versus structured pruning (middle and right panels).

**Unstructured pruning.** Unstructured pruning refers to the removal of fine-grained weights in contrast to a group of weights. It is the simplest and most sparsity-inducing type of pruning because trained neural networks are less sensitive to one weight than a specific block.

The most intuitive pruning scheme is to remove weights based on their absolute values, which is the simplest form of magnitude-based pruning, so it does not require any data. This simple approach has been studied early (Hagiwara, 1993) and is very effective (Han et al., 2016, Zhu & Gupta, 2017, Gale et al., 2019, Hoefler et al., 2021). In general, it involves re-training to adapt the model to its new architecture.

While there are a plethora of pruning algorithms, Gale et al. (2019) suggested that magnitude-based pruning provides state-of-the-art or comparable performance to other pruning methods (Thakker et al., 2020, Louizos et al., 2018b).

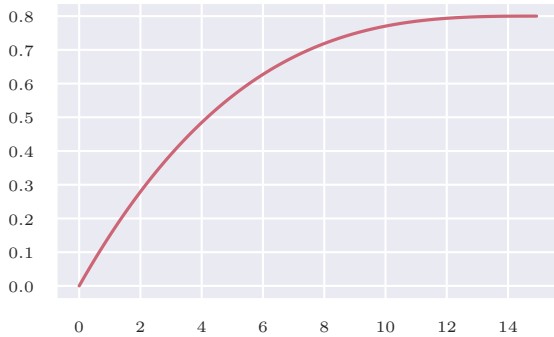

Figure 6: Pruning rate over epochs with a polynomial schedule function (Zhu & Gupta, 2017) with $s_f = 0.8$, $s_0 = 0$, $t_0 = 0$, $n = 13260$, $\Delta t = 100$ (Equation equation 6).

In particular, Zhu & Gupta (2017) introduced a gradual sparsity technique using a polynomial during the training schedule as follows

$$s_t = s_f + (s_0 - s_f)\left(1 - \frac{t - t_0}{n\Delta t}\right)^3, \quad \text{for } t \in \{t_0, t_0 + \Delta t, \ldots, t_0 + n\Delta t\}, \tag{6}$$

where $s_t$ and $t$ are the current sparsity and step, $s_f$ is the target sparsity, $s_0$ and $t_0$ are the initial sparsity and training step (usually 0), $n$ is the number of pruning steps, and $\Delta t$ is the pruning step frequency. In other words, at every $\Delta t$, a gradual number of weights is set to zero based on their magnitude until we reach the desired sparsity level. The objective of the polynomial schedule is to prune quickly and early when there is the most redundancy, and then slow down the pruning rate as there is little remaining redundancy (Zhu & Gupta, 2017).

Noting that pruning is a form of regularization, Golatkar et al. (2019) found that the early regularization phase is the most critical to performance and that late regularization can even worsen the results, thus supporting the effectiveness of polynomial schedule.

The advantage of magnitude-based pruning is that it is model- and task-agnostic, can seamlessly incorporate within training, and is easy to implement. Moreover, progressive pruning (Zhu & Gupta, 2017) is natively supported by the TensorFlow framework (Abadi, M. et al., 2016). Additionally, they demonstrate a 90% sparsity rate with acceptable accuracy loss and found that their approach on large-sparse networks performs better than their smaller-dense counterpart. An explanation of this is that larger models are easier to prune because the magnitude of single weights becomes smaller as the model grows larger when the model has converged (Neill, 2020). However, the biggest disadvantage is that unstructured pruning results in sporadically induced weights, which may be difficult to efficiently leverage on embedded hardware, but previous work demonstrated that it is possible to leverage high sparsity with practical encoding (Han et al., 2016).

**Structured pruning.** Structured pruning alters the architecture of the neural network in blocks, such as neurons, filters, or an entire row or column of a weight matrix. Structured pruning can be induced by using a systematic criterion based on redundancy, as in Srinivas & Babu (2015), where neurons were removed in neural networks by identifying duplicate pairs of neurons, performing a recovery step to compensate for removal. Another common approach is to use regularization penalty to encourage pruning at the channel level in CNN models (He et al., 2017, Liu et al., 2017), by neurons (Alvarez & Salzmann, 2016), or layers (Wen et al., 2016), resulting in models with 60% sparsity without significant loss. The clear advantage of structured pruning is that it is hardware efficient because it may allow skipping entire filters or rows during a matrix multiplication, as suggested in Figure 5. However, block-based pruning techniques have strict compression rules that make them more difficult

to achieve without degrading performance and require a certain amount of block sparsity to obtain a faster run time than baseline (Thakker et al., 2020). However, recent research suggests that wider and sparser networks generalize better than their smaller dense counterparts designed by structured pruning (Zhu & Gupta, 2017, Li et al., 2020, Golubeva et al., 2021, Timpl et al., 2022, Ballas, 2022).

**Pruning based on Bayesian methods.** Among all, Bayesian inference can be used to promote sparsity in the model. Bayesian methods provide the posterior distribution over the parameters of the model, given the dataset and a prior distribution. As a result, this posterior distribution encompasses more information than a simple vector of optimal parameters: variance of the parameters, thickness of their tails, etc. Besides, by tuning the prior distribution, the user can impose some structure to the posterior distribution, which can be used to encourage sparsity in the model.

A popular and intuitive prior is the *spike-and-slab* prior, introduced by Mitchell & Beauchamp (1988) and used in neural networks by Srinivas et al. (2017), Jantre et al. (2023), for instance. This prior is a mixture between a Dirac at 0 (the *spike*) and a distribution with a continuous density (the *slab*), e.g., a zero-mean Gaussian distribution. That way, the spike-and-slab prior pushes the parameters towards 0. More complex, the *Horseshoe* prior (Carvalho et al., 2009, Ghosh et al., 2019) has been designed to have an infinite density at 0 and Cauchy-like tails. Thus, the horseshoe prior encourages the parameters to be exactly 0 while allowing extreme values. Another regularization technique, the drop-out (Srivastava et al., 2014b), has led to the development of the *log-uniform* prior by Neklyudov et al. (2017). Although improper, this prior is designed to be agnostic about the order of magnitude of the parameters. As a result, its density tends to infinity at 0, so small values are encouraged. However, to make the log-uniform prior proper, it is common to set its density to 0 outside an interval spanning several orders of magnitude, as described below:

$$\text{Spike-and-slab:} \quad p(x) = p_0 \delta(x) + (1 - p_0) \frac{1}{\sqrt{2\pi\sigma_0^2}} \exp\left(-\frac{x^2}{2\sigma_0^2}\right), \ p_0 \in (0, 1), \ \sigma_0 > 0,$$

$$\text{Horseshoe:} \quad X_i \,|\, \lambda_i, \tau \sim \mathcal{N}(0, \lambda_i^2 \tau^2); \ \lambda_i \sim \mathcal{C}^+(0, a); \ \tau \sim \mathcal{C}^+(0, b), \ a > 0, \ b > 0, \quad (7)$$

$$\text{Proper log-uniform:} \quad p(x) = \frac{1}{2|x| \log(b/a)} \mathbb{1}_{[a,b]}(|x|), \ 0 < a < b,$$

where $\mathcal{C}^+(0, a)$ is the half-Cauchy distribution with scale parameter $a$. These densities are illustrated in Figure 7.

Beyond the choice of the prior, one should pay attention to the choice of the *approximate* Bayesian method and the search space of the approximate posterior. In fact, it is usually too costly to compute the exact posterior distribution of the parameters of large models such as neural networks (Arbel et al., 2023, Papamarkou et al., 2024). Therefore, one has to choose an approximate Bayesian method and a search space of the posterior distribution. For instance, it is common to use *variational inference* (Graves, 2011) and look for an approximate posterior consisting of independent Gaussian distributions over the set of parameters (where their mean and variance are trained). In Srinivas et al. (2017), the candidate posterior distributions for one parameter $\theta$ are the mixtures between the Dirac at 0 and $\mathcal{N}(\mu, \sigma^2)$, with mixture parameter $g$: the trained parameters are then $g$, $\mu$, $\sigma$. In that case, the value of $g$ is directly related to the sparsity: if $g = 0$, then $\theta = 0$, so $\theta$ can be pruned.

**Summary.** In summary, pruning has strong theoretical and practical incentives that make it a high-potential and relevant choice. Unstructured pruning approaches are more flexible across diverse architectures and yield the highest sparsity rate, while structured pruning approaches are more hardware efficient.

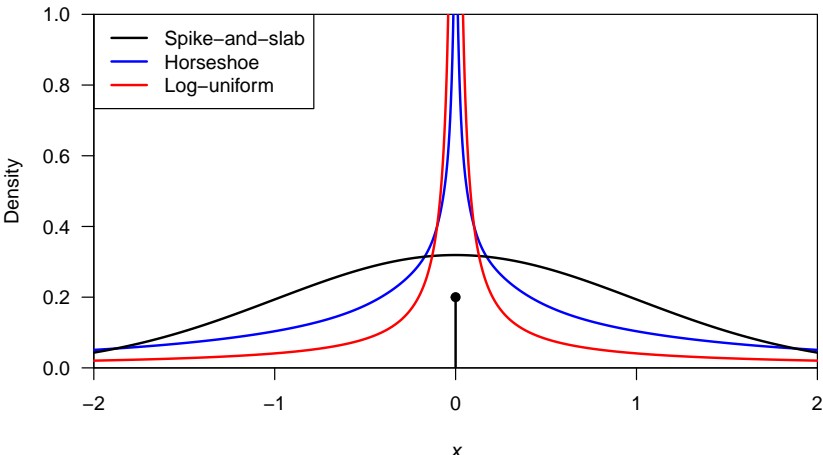

Figure 7: Prior densities promoting sparsity as described in Equation equation 7, illustrated with the following hyperparameters: Spike-and-slab with $p_0 = 0.2$ and $\sigma_0 = 1$; Horseshoe with $a = b = 1$; Proper log-uniform with $a = 10^{-5}$ and $b = 2$.

Moreover, multiple works have shown that combining pruning with other model compression methods, such as quantization, can produce a high compression rate without significant performance loss (Han et al., 2016, Van Baalen et al., 2020, Zhang et al., 2021b).

### 4.3 Quantization

A different perspective on model compression is quantization. It is a method that maps input values from a larger set (often continuous) to a smaller set (often discrete) (Gholami et al., 2022) to find lossless approximations of numerical input values, and can be seen in related work dating back to the 1800s in the foundations of calculus (e.g., least-squares, approximation of integrals) (Gray & Neuhoff, 1998, Gholami et al., 2022).

In particular, fixed-point attempts to represent continuous values (larger set) with a fixed amount of precision (smaller set); thus, quantization is a mandatory method to meet the low-power requirements of fixed-point arithmetic inference on MCUs, as stated in Section 3.3.

Recent work on neural network quantization builds upon prior work but presents unique challenges due to the high power footprint and overparameterized nature of deep learning models. The inherent redundancy in deep learning models allows for some leniency in quantization errors, limiting accuracy loss (Guo, 2018, Gholami et al., 2022). Consequently, very small models, that can be found in TinyML, should be more sensitive to quantization.

Minimizing quantization performance loss can be seen as an optimization problem, where the objective is to find a discrete distribution (quantized weights) that is closest to the original distribution (real weights, activation, or data). In practice, this translates by rounding or truncating the model's parameters (weights, activations) and data from floating points (e.g., 32-bits) to integer values (e.g., 8-bits).

Compared to pruning, quantization often results in less accuracy loss because weights lose precision but are not removed, hence a lower level of information loss (Saha et al., 2022).

Quantization approaches can be characterized by several factors: the stage of the quantization process as quantization-aware training (QAT) or post-training quantization (PTQ), the type of quantization steps as uniform or non-uniform, and the arrangement of quantization levels around the zero-point $Z$ as symmetric or asymmetric (Equation equation 8).

**Quantization-aware training (QAT).** QAT involves integrating quantization into the training process or fine-tuning the model by simulating the effects of quantization during the forward or backward pass. However, the quantized function is not differentiable (Equation equation 8) and can result in zero-gradients in low bit-precisions, making it difficult to train the model. Prior works have quantized values in the forward pass and used real values during the backward pass such as the straight-through estimator (STE) (Bengio et al., 2013, Courbariaux et al., 2015, Jacob et al., 2018), or other approximations (Yin et al., 2018, Louizos et al., 2018a). In addition, Choi et al. (2018) learns to optimize the range of activation clipping values and then linearly quantize both weights and activations to 4-bits, while Bhalgat et al. (2020) uses a gradient estimate to learn scaling factors of weights and activations. Alternatively, Darabi et al. (2018) employ regularization to force the weights to converge to binary values during training, which is generalized in Lê et al. (2023) to any bit-precision and using a schedule for progressive quantization during training. The objective of QAT is to obtain a stabilized quantized model by the end of training. These methods enable below 8-bit quantization and even down to 1-bit weights or activations (Courbariaux et al., 2015, Rastegari et al., 2016, Hubara et al., 2016, Liu et al., 2018a, Qin et al., 2020) with competitive results compared to full precision networks and PTQ. Additionally, AskariHemmat et al. (2022) found that quantization is a form of regularization, where the induced quantization noise can help improve generalization, and particularly to 8-bits on several computer vision tasks. However, QAT often requires a lot of tuning, additional computation, and access to the dataset to re-train the model, especially for low-bit quantization.

**Post-training quantization (PTQ).** PTQ is the simplest and fastest approach, where quantization can be applied to any trained model without re-training or access to the dataset (Han et al., 2016, Choukroun et al., 2019, Banner et al., 2019, Cai et al., 2020, Fang et al., 2020). Previous work corrected the mean and variance of quantized weights (Banner et al., 2019, Gholami et al., 2022), or minimized the mean squared error between the quantized and full-precision distributions (Choukroun et al., 2019), allowing 4-bit quantization with acceptable performance. Another approach used piecewise linear functions to partition the quantization range into non-overlapping regions for each weight in order to minimize the quantization error (Fang et al., 2020).

The most widely used quantization method for MCUs is uniform affine PTQ to int8 because it is straightforward and supported by MCUs (Krishnamoorthi, 2018, Gholami et al., 2022, Saha et al., 2022). Moreover, uniform PTQ with int8 provides sufficient performance compared to the original full-precision 32-bit (FP32) model for a wide variety of NNs (Krishnamoorthi, 2018, Lee et al., 2018, Fang et al., 2020). However, PTQ may lead to a more significant loss in accuracy, especially for quantization below 8 bits (Banner et al., 2019, Gholami et al., 2022).

**(Non-)uniform quantization.** In uniform quantization, the quantization steps are evenly spaced, so it is the most straightforward type of quantization while being natively supported in all embedded hardware (Saha et al., 2022).

In contrast, non-uniform quantization may better capture the original distribution, thus yielding higher accuracy (Gholami et al., 2022). For example, Miyashita et al. (2016), Zhou et al. (2017) uses a logarithmic distribution with exponential quantized steps instead of linear steps. Alternatively, Fu et al. (2020) quantize activations and gradients by finding optimal quantization points that fit their full-precision distributions based on their Weibull prior properties (Vladimirova et al., 2019; 2021), and obtained competitive results compared to the full precision training using less bits than their uniform-based counterpart (Fu et al., 2020).

However, non-uniform quantization schemes are challenging to deploy on embedded hardware because they require a custom implementation to efficiently exploit their specific distribution, in contrast to uniform quantization which is deployable out of the box. Therefore, we restrict the scope of our review to uniform quantization schemes for a wide hardware support.

**(A-)symmetric quantization.** In symmetric quantization, the lower and upper bounds of the quantization range are equidistant from the zero-point, and $Z = 0$, which simplifies as follows

$$Q(r) = \text{int}(r/S) - Z, \tag{8}$$

where $Q$ is the quantization function, $r$ the value to quantize, $S$ a scaling factor, $Z$ represents the zero-point value in the integer discrete space, $\alpha, \beta$ denote the lower and upper bounds ($\alpha < \beta$), respectively, of the clipping range where we constrain $r$, and $b$ is the bit-width.

The scaling factor for symmetric and asymmetric quantization is computed as follows:

$$\begin{aligned} S_{\text{sym}} &= \frac{\max(|\alpha|, |\beta|)}{2^{b-1} - 1} \\ S_{\text{asym}} &= \frac{\max(|\alpha|, |\beta|)}{(2^b - 1)/2}, \end{aligned} \tag{9}$$

Asymmetric quantization schemes consider the full range of quantized values, e.g., $[-128, +127]$, in contrast to $[-127, +127]$. This provides a slightly larger range to minimize quantization error but is a more complicated implementation due to the zero point $Z \neq 0$ in Equation equation 8, and may lead to more computational overhead (Wu et al., 2020).

**Quantization based on Bayesian methods.** Similarly to the case of pruning, Bayesian inference can be used to reduce the number of bits necessary to encode a continuous parameter. For instance, Van Baalen et al. (2020) have proposed a method in the variational inference framework (Graves, 2011): each parameter of a neural network is decomposed as a sum of gated residuals:

$$x = z_2(x_2 + z_4(\epsilon_4 + z_8(\epsilon_8 + z_{16}(\epsilon_{16} + z_{32}\epsilon_{32})))),$$

where $x_2$ is the basic 2-bits approximation of $x$, the $\epsilon_n$ are the $n$-bits residuals of $x$, and the $z_i$ are the corresponding gates. In this example, $x$ is allowed to be pruned or approximated on $2^n$-bits for $n \in \{1, 2, 3, 4, 5\}$. The $(z_i)_i$ are dependent Bernoulli random variables whose parameters are trained: if all $z_i$ tend to become 0, then $x$ can be pruned; if $z_2$ tend to be always 1 and the others 0, then $x$ can be efficiently approximated by its 2-bits part; if all $z_i$ tend to be always 1, then $x$ should remain coded on 32 bits. In this setup, the optimal level of quantization (in a Bayesian sense) is discovered progressively during training and can be heterogeneous across the parameters. Moreover, the allowed quantization levels span a large interval, from the usual 32-bits quantization to pruning.

Also, the entire posterior distribution provided by Bayesian inference can be used to improve quantization methods. For instance, Yang et al. (2020) have developed a quantization method that can be applied to a model for which a posterior distribution is already known for each of its parameters. In this work, the posterior distribution of each parameter is transformed by a function, which is the CDF of the prior distribution. Then, the mode of the resulting function is quantized with precision depending on its width: if the mode has a large width, then a few bits are necessary to encode it. With this setup, the partition used for quantization is, at least, adapted to the prior distribution, and leads to a more efficient quantization when applied to posterior distributions. Finally, Meng et al. (2020) trains binary neural networks using the Bayesian learning rule Khan & Rue (2023), an algorithm inspired by the Bayesian paradigm. This approach enables uncertainty quantification while providing state-of-the-art results.

**Summary.** In summary, quantization methods have a long history and exist in many flavours to achieve lossless approximations in the most constrained settings. QAT emerges as a superior option in below 8-bit settings, but is more complex and requires more computations than PTQ.

However, uniform PTQ with lower bit quantization is more sensitive due to the distributional properties of weight, which are clustered around zero (Gaussian or Laplacian) (Han et al., 2016, Lin et al., 2016),

and few of them are in a long tail (Sub-Weibull) (Vladimirova et al., 2019; 2021). Consequently, uniform quantization maps too few quantization levels to small weights and too many to large ones, leading to performance loss (Fang et al., 2020). However, overparameterized models are less sensitive to PTQ due to having more degrees of freedom (Neill, 2020) in contrast to smaller models.

Thus, we would favour uniform 8-bit PTQ due to its simplicity and acceptable results until we need lower-bit precision for more power footprint reduction.

## 4.4 Weight-sharing

Weight-sharing is the simplest form of model compression, involving sharing weights values in different parts of the model, so it imposes a model architecture prior to training (Neill, 2020). We could set the amount and location of weight-sharing in a strategic way in the model, such as in rows or columns of the weight matrix, for efficient inference. However, manual weight-sharing design may be difficult because we cannot predict the final performance, even if redundancy is part of the design of deep learning models.

Prior works have used an automated approach, such as clustering weights with K-means that shares the centroid value among weight clusters with re-training (Wu et al., 2018), where they compressed a CNN model by a factor of three without significant loss, or by using a penalty term to encourage grouping weight (Nowlan & Hinton, 1992, Ullrich et al., 2016).

In particular, quantization is a form of weight-sharing because lowering the bit-precision of parameters forces them to be aggregated into a common set of values.

## 4.5 Low-rank matrix and tensor decompositions

Since neural network weight parameters are essentially matrix or tensors, we can apply approximation methods from linear algebra such as single value decomposition (SVD) or its generalization to tensor decomposition (TD) (Neill, 2020). The weight matrix is then replaced by a product of two lower-rank matrices (Xue et al., 2013, Sainath et al., 2013, Novikov et al., 2015, Alvarez & Salzmann, 2017). In particular, Alvarez & Salzmann (2017) obtained a compression rate of up to 96% compared to the original model.

However, these methods require additional hyperparameter tuning (Lebedev et al., 2015), as well as trial and error to find the optimal rank, which may not generalize between applications. Furthermore, for MCUs, it is crucial to consider that the incorporation of additional products from the lower rank matrix may not always lead to increased efficiency and reduced power consumption, so further evaluation of the device is required.

## 4.6 Summary

In summary, we have provided a comprehensive overview of the key methods to design and train efficient TinyML models, accompanied by their related theoretical concepts and practical implications. These methods have generated growing interest, as they bridge the gap between deep learning theory and the deployment of efficient neural networks.

Specifically, model pruning, knowledge distillation, and quantization have demonstrated very promising compression rates, particularly in larger-scaled networks (Mobile or Cloud size) that are more robust to model adjustments. Furthermore, some model compression methods are also forms of regularization that can even help the model to generalize better. Thus, these approaches show high potential to meet the ultra-low-power requirements MCUs.

In practice, since TinyML is at an early stage, tools and processes are not mature enough yet to evaluate and truly leverage the high compression rate of existing methods for ultra-low-power MCUs,



Figure 8: TinyMLOps pipeline.

so we will review practical TinyML tools and aspects of the deployment of compressed neural networks in the next section.

# 5 Deploying deep learning models on ultra-low-power MCUs

In this section, we define and review existing tools for the end-to-end deployment of efficient neural networks on ultra-low-power MCUs.

**TinyMLOps.** The first framework for training deep learning models was developed in 2008 (Theano Development Team, 2016), with TensorFlow (Abadi, M. et al., 2016) and PyTorch (Paszke et al., 2019) following suit in 2015 and 2016, respectively. These frameworks enabled the large-scale development and deployment of deep learning models, which in turn led to the emergence of Machine Learning Operations (MLOps) (Kreuzberger et al., 2022). MLOps consolidates best practices and outlines steps for mitigating technical debt (Sculley et al., 2015) during the development and deployment of machine learning systems.

In contrast, the earliest known publication on TinyML dates back to 2019 (Han & Siebert, 2022), and the first dedicated deep learning framework for microcontrollers, TensorFlow Lite for Microcontrollers (TFLM), was also released in 2019 (Warden & Situnayake, 2020, David et al., 2021). As TinyML gained traction in the industry, MLOps naturally expanded to include TinyMLOps as a subset (Sah et al., 2022, Leroux et al., 2022, Lê & Arbel, 2023), focusing on refining the process of deploying machine learning on embedded devices, as depicted in Figure 8. In the context of TinyML, deployment refers to the process of taking a trained model and enabling it to run on an embedded system, such as compiling the model, firmware integration, and verification of the solution on the target device.

Consequently, the TinyMLOps ecosystem is still in an earlier stage than MLOps, with challenges yet to be fully addressed. We detail here the challenges faced by TinyMLOps tools, as well as existing solutions.

## 5.1 Challenges for TinyML tools

The fundamental characteristic of TinyML is the tight dependency between software and hardware components. In fact, failure to adapt the delivered machine learning software to the constraints of particular hardware renders it unusable, resulting in wasted efforts in previous TinyMLOps steps. Additionally, the diverse landscape of embedded hardware further complicates the task of developing a versatile software base capable of supporting a wide range of embedded hardware platforms (Sah et al., 2022, Leroux et al., 2022), resulting in a manual and iterative approach to the design of new models. As a result, designing new models that work on different hardware remains a manual and iterative approach (different firmware, debugging interfaces...). The challenge of TinyMLOps is to improve the entire pipeline, from design to deployment, from data to computation.

Even though TinyML shares some tools with traditional ML (e.g., TensorFlow, PyTorch, Tensorboard), its more recent emergence means that specialized tools are not yet created or are less mature in providing comprehensive solutions. As the TinyML community continues to grow, greater awareness and adoption of tools will lead to faster innovation and the development of comprehensive solutions.

## 5.2 TinyML tools solutions

We restrict TinyML frameworks to the one that supports TensorFlow models as input due to its wide adoption in the industry and that also targets Arm Cortex-M MCUs for inference.

We essentially consider these two common approaches to TinyML frameworks (Sipola et al., 2022):

1. Using a runtime that loads the model from read-only device memory at runtime (e.g., Tensor-Flow Lite Micro),

2. Using a transcompiler that converts and compiles models to C or C++ code that then can be built within a project (NNoM, Edge Impulse, $\mu$TVM).

### 5.2.1 Low-level library

**CMSIS-NN.** CMSIS-NN (Cortex Microcontroller Software Interface Standard for Neural Networks) is a low-level library specifically developed by Arm (Lai et al., 2018) for the Cortex-M microcontroller ecosystem (Table 4). It provides a collection of efficient neural network core functions for low-level acceleration. These functions include optimized operations for common neural network operations, such as fully-connected (FC) layers, convolutions, and activation functions (ReLU, sigmoid, tanh...). CMSIS-NN has been shown to provide a 4.6x speedup and 4.9x energy savings over non-optimized convolutional models (Lai et al., 2018, Saha et al., 2022).

### 5.2.2 TinyML frameworks

**TensorFlow Lite Micro (TFLM).** This framework is an extension of the TensorFlow ecosystem, specifically designed for deploying neural networks on low-power MCUs such as ARM Cortex-M (David et al., 2021, Warden & Situnayake, 2020). (Sipola et al., 2022, Ray, 2022, Saha et al., 2022). TFLM emphasizes portability by discarding uncommon features, data types, and operations and avoids reliance on specialized libraries or operating systems, thereby achieving memory efficiency and support for a wide range of hardware. It converts and quantizes a 32-bit floating-point TensorFlow model to a compressed flat buffer file (.tflite) using 8-bit integers for weights and 32-bit integers for activations and data. TFLM uses an interpreter-based approach to process the neural network graph at runtime and consists of three primary components: operator resolver, memory stack pre-allocation, and interpreter (Sponner et al., 2021, Schizas et al., 2022). The operator resolver links only essential operations to the model binary file, and the memory stack is used for initialization and storing runtime variables. The interpreter resolves the network graph at runtime, allocates the memory stack, and performs runtime calculations. More technical details are provided in David et al. (2021), Schizas et al. (2022).

However, TFLM has limitations, such as missing support of some layers or operations (GRU, Conv1D, some important activation functions...), arbitrary bit-widths of weights, and activations. Moreover, TFLM lacks target-specific optimizations during compilation because it relies on a graph-level representation that does not include device-specific function kernels and execution details (Sponner et al., 2021, Schizas et al., 2022), and can result in larger memory usage, so it may not meet our extreme memory requirements. Moreover, it does not provide built-in tools to measure power footprint metrics such as inference time or memory usage. Moreover, the interpreter-based approach at runtime makes it difficult to debug and extend, compared to standard compiled code, which hinders research efforts. Despite these limitations, TFLM remains the most popular choice for microcontroller-based deep learning applications.

**Neural Network on Microcontroller (NNoM).** This open-source framework (Ma, 2020) relies on a C code generation approach with a set of function calls. It is flexible, easy to debug, and supports a wide range of MCUs, but only supports models created using TensorFlow. The project includes a compiler that converts and quantizes a TensorFlow model to plain C code with 8-bit weights and

32-bit activations and data. Additionally, the NNoM compiler supports the CMSIS-NN to generate optimized code for ARM Cortex-M processors (Sipola et al., 2022). It does support all RNN layers including GRU, in contrast to TensorFlow. However, it does not support lower bit-width quantization and has a smaller community and adoption compared to TFLM, so this hinders the development of new features.

**Edge Impulse.** Lastly, Edge Impulse (Janapa Reddi et al., 2023) is a closed-source cloud service that develops TinyML machine learning models for edge devices and supports AutoML for mobile and microcontrollers (Saha et al., 2022, Ray, 2022). Edge Impulse provides a complete end-to-end model deployment solution, including data collection, feature extraction, training, and deployment (Saha et al., 2022), with an intuitive graphical interface and a friendly no-code approach. The training is carried out in the cloud and the learned model can be exported to an edge device using a data-forwarding capable connection (Schizas et al., 2022).

For model deployment, Edge Impulse uses an interpreter-less edge-optimized neural compiler, which directly compiles the model into C++ source code. This approach eliminates the need to store unused ML operators, resulting in reduced memory requirements at the expense of portability compared to TFLM. Studies have shown that the EON compiler can run the same model with 25%–55% less SRAM and 35% less flash memory than TFLM (Saha et al., 2022).

In conclusion, TinyML brings together the embedded systems and machine learning communities, which have traditionally operated independently. Both academia and industry have developed several software frameworks for TinyML to streamline the deployment of machine learning models on microcontrollers. In particular, we are interested in TFLM because it integrates with TensorFlow and provides a complete toolchain for deploying low-power models MCUs. We are also interested in NNoM because it provides a flexible and simple approach to quantizing and deploying models from plain C code and CMSIS-NN support for Arm Cortex-M MCUs. Moreover, these two frameworks are open-source, which makes them accessible as well as potentially extendable. However, these frameworks are still in the early stages of development, with some missing features and functionality. Despite their limitations, the current first generation of TinyML tools can transition the state-of-the-art machine learning models to ultra-low-power environments.

## 6   Limitations of TinyML

In this section, we systematically assess the limitations of current TinyML models when applied to standard datasets, with a specific focus on their memory size. Our primary objective is to identify the most efficient models that strike the optimal balance between performance and memory usage. By doing so, we aim to offer valuable insights to researchers and industry professionals, shedding light on the scale of TinyML models. Furthermore, our analysis aims to pinpoint the most suitable models among widely adopted options and various hardware platforms for their respective applications.

We focus here on the most common datasets found in TinyML model benchmarks for the following three tasks, see Figure 9 presenting accuracy against (flash) model size:

- *Image Classification*: **MNIST** is a basic dataset for image classification of handwritten digits. We also use **ImageNet**, a more challenging image classification dataset than MNIST due to its larger and more diverse images and labels, thus requiring more complex models.

- *Image Recognition*: **Visual Wake Word** is focused on the visual presence recognition of a person or an object in images.

- *Speech Recognition*: **Google Speech Commands v2-12** consists of short audio clips of spoken word commands with 12 classes to recognize.

**MNIST.** For MNIST, we find that $\mu$NAS (Liberis et al., 2021) clearly offers the best size-accuracy tradeoff and is below the Cortex M0+ memory limitation. The large LeNet (Han et al., 2016) has slightly better accuracy but is over the memory threshold. Then, the two versions of Sparse CNN (Fedorov et al., 2019) are both below the extreme low-power threshold, but their accuracies are still lower than $\mu$NAS. However, ProtoNN (Gupta et al., 2017) and Bonsai (Kumar et al., 2017) display the least favorable tradeoff, but ProtoNN is below the Cortex M0+ threshold.

**ImageNet.** We observe that ImageNet models require the largest models of all studied here, mostly above the ultra-low-power microcontrollers (Cortex M4 and M7) threshold. In particular, the large MCUNet (Lin et al., 2020) has the best accuracy tradeoff and is right below the Cortex M7 memory threshold. Both versions of SqueezeNet (Iandola et al., 2016) and MNasNet (Tan et al., 2019) have low accuracy, so they are unsuitable for practical application.

**Visual Wake Word (VWW).** We notice that no models are below the Cortex M0+ memory threshold, but the size of the RaScaNet models (Yoo et al., 2021) shows that it would be reachable with further research. In the ultra-low-power range, MSNet (Cheng et al., 2019) clearly provides the optimal size-performance tradeoff, but one could deploy the large RaScaNet for even lower power and acceptable accuracy. In comparison, the performance of MNasNet is less favorable. We also see that MobileNetV1 (Howard et al., 2017, Banbury et al., 2021b) and MicroNet (Banbury et al., 2021b) display the worst size-performance tradeoff.

**Google Speech Commands v2-12.** In the extremely low-power range, we note that FastGRNN (Kusupati et al., 2018) offers the best tradeoff, while in the ultra-low-power range, $\mu$NAS displays once again the best tradeoff. ConvGRU 4-bits (Lê et al., 2023), ShallowRNN (Dennis et al., 2019), Hello Edge DS-CNN (Zhang et al., 2018), TinySpeech-Z (Wong et al., 2020), LSTM-KP (Thakker et al., 2021), LMU-4 (Blouw et al., 2020), FastRNN (Kusupati et al., 2018), DS-CNN (Banbury et al., 2021a) and all have acceptable performance, but are still less favorable than $\mu$NAS. In contrast, all versions of MicroNet present the least optimal performance once more, where the large MicroNet is even above the Cortex M4 threshold.

**Summary.** Among the standard datasets, TinyML models are able to comply with extreme-low power constraints as low as 8 kB for a speech recognition task and a simple image classification dataset, with a given tradeoff on accuracy. In this regard, further research efforts are required for an image recognition task and a more complex image classification problem. Otherwise, Cortex M4 is sufficient to run most models for all tasks with the best accuracy.

The industrial cost of exceeding a hardware memory threshold is high. Emphasizing the successful deployment of models on the most constrained microcontrollers is crucial, given the substantial economic impact. Even though microcontroller power classes (extreme low-power and ultra-low-power) have minor price differences (ranging from 1 to 3 dollars) and are inexpensive, the price significance magnifies when considering the billions of annual unit market sales, resulting itself in billions of yearly savings. Thus, designing efficient models is critical for the TinyML industry, and inherently comes with a price tradeoff.

## 7 Conclusion and discussion

**Summary.** In Section 2 we presented the state of neural networks and motivated our interest in them for our applications, then we provided an overview of MEMS-based applications, emphasized the opportunities and challenges of our extremely low-power constraints, that reinforce the need for more TinyML research efforts in Section 3. Then in Section 4, we presented the existing methods to design efficient neural networks on ultra-low-power MCUs, and provided an overview of existing tools

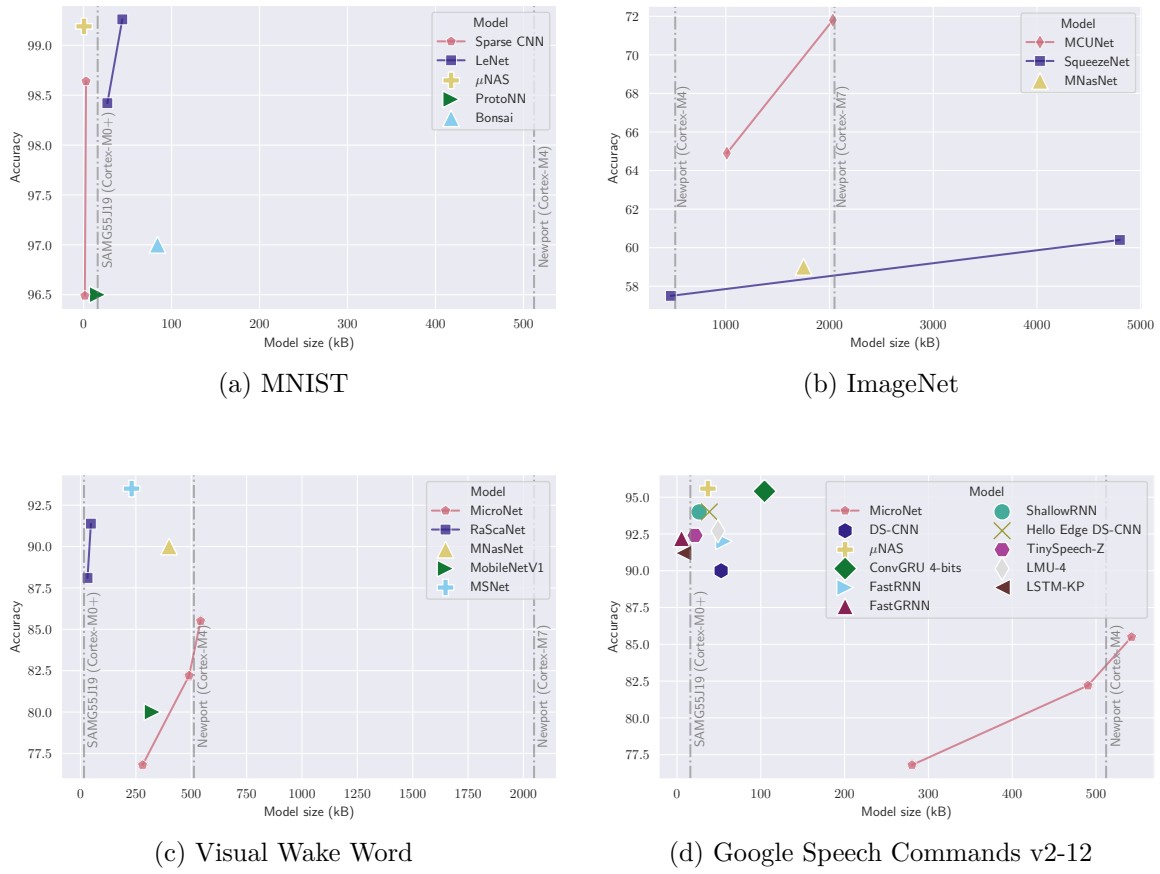

(a) MNIST

(b) ImageNet

(c) Visual Wake Word

(d) Google Speech Commands v2-12

Figure 9: Flash model size versus accuracy on the four considered datasets. Vertical grey dashed lines indicate hardware storage limits for Cortex-M0+, Cortex-M4 and Cortex-M7 (see Table 4).

to deploy neural networks to enable for TinyML applications in Section 5. Finally, we examined the current limitations in the field of TinyML in Section 6.

**Open challenges.** TinyML is faced with a number of open challenges. Ensuring the robustness of TinyML models against *adversarial attacks* remains a significant challenge. Adversarial attacks can manipulate input data to mislead the model, posing security risks in critical applications. Research is needed to develop robust TinyML models that can withstand various forms of adversarial attacks. This includes exploring techniques such as adversarial training, input perturbation defences, and understanding the trade-offs between model complexity and robustness. Additionally, many edge devices in TinyML applications operate in *dynamic environments* with fluctuating resource availability. Managing resources such as power, memory, and bandwidth dynamically to adapt to changing conditions is a complex challenge. Further investigation of adaptive resource management strategies for TinyML models will be required in the future, considering real-time changes in resource availability. This includes exploring techniques for dynamic model adaptation, on-the-fly optimization, and resource-aware scheduling to ensure optimal performance under varying conditions. Addressing these challenges would not only enhance the robustness and adaptability of TinyML models but also contribute to the broader applicability of TinyML in diverse and dynamic edge computing environments.

**Emerging trends and technologies.** Several trends and technologies may impact the field of TinyML in the coming years. *Edge AI and Edge Computing*: the integration of TinyML with edge computing is a prominent trend, enabling the processing of machine learning models closer to the data source. This approach reduces latency, addresses bandwidth constraints, and optimizes TinyML models for resource-constrained edge devices. *Quantum Computing*: quantum computing holds the potential to revolutionize the field of TinyML by accelerating model training and optimization processes. As quantum computing technologies mature, researchers may explore their application to enhance the efficiency and performance of TinyML models. *Custom Hardware Accelerators*: the development of custom hardware accelerators designed for efficient execution of TinyML models on edge devices is a key trend. Specialized hardware architectures aim to improve both performance and energy efficiency, contributing to the widespread deployment of TinyML in diverse applications.

These trends collectively signify a shift towards more efficient, decentralized, and specialized computing approaches, paving the way for advancements in the deployment and optimization of TinyML models on resource-constrained devices at the edge. They suggest a dynamic landscape for the future of TinyML, with innovations in hardware, communication, and algorithmic approaches contributing to the continued evolution of this field. Researchers and practitioners in TinyML should stay informed about these trends to harness their potential benefits and address new challenges.

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
