# OpenReview forum: "Efficient Neural Networks for Tiny Machine Learning: A Comprehensive Review"
_TMLR — Withdrawn by Authors_

### Review · Reviewer_EFB4 · 2024-03-29

**Summary Of Contributions:**

This is a literature review paper targetting the use of TinyML techniques on ultra-low-power MCU applications. The paper provides an accessible and comprehensive guide to TinyML, introducing NN design, MCU application and system constraints, efficient deep leanring methods, system deployment, and summarize the current state-of-the-art and limitations in the field of TinyML.

**Audience:**

Yes

**Broader Impact Concerns:**

No concern on the boarder impact.

**Claims And Evidence:**

Yes

**Requested Changes:**

1. Reduce the length of introducing different neural network models, remove parts in Sec. 2.2 and Sec. 2.3 that are not relavent to the scope of TinyML
2. Revise the organization of Sec. 4, include seperated subsection for efficient model architecture design and neural architecture search
3. Discuss the impact of differnet techniques in Sec. 4 and 5 on MCU resource
4. Make better transition from Sec 4 and 5 to Sec 6. Some of te identified "optimal solution" mentioned in Sec 6 are not introduced in prevous sections, and the path forward is unclear despite mentioning multiple limitations.

**Strengths And Weaknesses:**

## Strength

1. This paper provides a full-stack prespective of TinyML, covering application, deep learning models, efificency methods, system, and hardware support. The coverage is good for the interest of both researchers and developers.
2. The paper pay special focus to the MCU application, introducing the resource constraints of common MCU devices (Sec 3) and software and system frameworks for deploying deep learning methods on MCU devices (sec 5). These are great addition over previous review papers on the TinyML topic.

## Weakness

1. This paper spends some efforts on making general introduction to neural network in Sec. 2. Though I see the intention, I think having such introduction may be redundant in an review paper for TinyML devices as there have been plenty of better resources introducing NNs out there, especially for the contents in Sec. 2.2 and 2.3. More over, these sections are not detailed enough for people without any knowledge to understand. Maybe it's better to introduce readers to related summary work on NN models rather than having a long section introducing NN itself.
2. The organization of Sec. 4 is questionable. A discussion of efficient RNN is strangely placed. Meanwhile, the techniques mentioned in the rest of the section is also appliable to RNN. Baysian methods are specifically pointed out in 4.2 and 4.3 without much reasoning on why they need special attention.
3. The coverage of references in Sec. 4 is inadequate. For example, Sec 4.2 totally overlooks the study on different pruning criteria beyond magnitude, such as Taylor sensitivity and Hessian etc. Also regularization-based pruning would worth specific discussion.
4. Each subsection in Sec 4 mainly lists a bunch of methods, without clearly discussing each of their contributions and limitations. Moreover, the principle concepts of pruning, quantization, and low rank are not introduced. How these techniques contributes exactly to MCU applications and how they compare against each other are also not discussed.
5. Efficient model design (squeezeNet, mobileNet) etc. and Neural architecture search are also promising and important directions of TinyML, yet they are totally overlooked in Sec. 4
6. The transition from method introduction in Sec 4/5 to the limitation in Sec 6 is awkward. Several work mentioned in Sec. 6 are not discussed in previous sections, and the connection between the techniques introduced and the limitations observed is not clear.
7. No insight is provided on how to move forward given the current limitations

---

### Review · Reviewer_8AUV · 2024-03-31

**Summary Of Contributions:**

This paper is a review of the advancements in efficient neural networks and the deployment of deep learning models on ultra-low-power microcontrollers (MCUs) for TinyML applications. The review includes a summary of efficient neural networks, deployment tools, and the current limitations of TinyML.

**Audience:**

Yes

**Broader Impact Concerns:**

No concerns about the ethical implications have been raised.

**Claims And Evidence:**

No

**Requested Changes:**

1. Summarize the key takeaways/insights in the introduction section.
2. Explicitly list the contributions over the prior TinyML reviews instead of using handwaving statements, e.g., "Unlike these works, we provide an accessible and comprehensive guide to TinyML, aimed at both deep learning practitioners and developers working with ultra-low-power devices interested in integrating neural networks into microcontrollers."
3. Simplify the reviewed techniques to only include those designed specifically for TinyML applications.
4. Add more experimental details about Figure 9 and include additional efficiency metrics other than memory, or cite prior works to justify that model size is currently the main focus of TinyML.

**Strengths And Weaknesses:**

Strengths:
1. The need for TinyML is well-motivated.
2. The summary of the different resource requirements for TinyML compared to ML on the cloud and other devices is very clear and helps readers understand the key challenges in TinyML.
3. The summary of existing tools for TinyML deployment can help readers know where to start when working on TinyML projects.

Weaknesses:
1. The contributions are unclear compared to prior reviews on TinyML [1-7]. Specifically, the author claims that "Schizas et al. (2022) and Ray (2022) give a detailed overview of applications, hardware, and deployment processes for TinyML, without introducing neural networks." As reviews on TinyML, there is no need to explicitly introduce neural networks, as there are already many reviews and tutorials on neural networks. Thus, I think the author needs to elaborate more on the claim that "unlike these works, we provide an accessible and comprehensive guide to TinyML, aimed at both deep learning practitioners and developers working with ultra-low-power devices interested in integrating neural networks into microcontrollers."

2. The model compression techniques introduced in Sec. 4 are not dedicated to TinyML. From Sec. 4.1 to 4.3, the reviewed techniques are not specifically designed for TinyML, and the author did not mention the unique challenges of applying those techniques to TinyML applications. The general review of pruning, quantization, and distillation can already be found in many existing reviews. Thus, the contribution of this part is unclear.

3. There are no justifications for the model selection when discussing the limitations of TinyML. In Figure 9, there is no justification for the dataset selection, nor is there any justification for whether the benchmarked models are the state-of-the-art (SOTA) TinyML models. Also, although memory allocation is one of the key factors to consider when running TinyML models, latency is also an important factor. As pointed out in [8], models with smaller sizes may be slower in terms of latency. Thus, it would provide insights by adding an evaluation in terms of latency.

[1] Ray, Partha Pratim. "A review on TinyML: State-of-the-art and prospects." Journal of King Saud University-Computer and Information Sciences 34.4 (2022): 1595-1623.

[2] Dutta, Lachit, and Swapna Bharali. "Tinyml meets iot: A comprehensive survey." Internet of Things 16 (2021): 100461.

[3] Han, Hui, and Julien Siebert. "TinyML: A systematic review and synthesis of existing research." 2022 International Conference on Artificial Intelligence in Information and Communication (ICAIIC). IEEE, 2022.

[4] Schizas, Nikolaos, et al. "TinyML for ultra-low power AI and large scale IoT deployments: A systematic review." Future Internet 14.12 (2022): 363.

[5] Abadade, Youssef, et al. "A comprehensive survey on tinyml." IEEE Access (2023).

[6] Saha, Swapnil Sayan, Sandeep Singh Sandha, and Mani Srivastava. "Machine learning for microcontroller-class hardware: A review." IEEE Sensors Journal 22.22 (2022): 21362-21390.

[7] Kallimani, Rakhee, et al. "TinyML: Tools, applications, challenges, and future research directions." Multimedia Tools and Applications (2023): 1-31.

[8] Gholami, Amir, et al. "Squeezenext: Hardware-aware neural network design." Proceedings of the IEEE conference on computer vision and pattern recognition workshops. 2018.

---

### Review · Reviewer_xhdn · 2024-04-03

**Summary Of Contributions:**

This paper reviews advancements in optimization and deployment of machine learning models on ultra-low-power microcontroller units (MCUs) for TinyML applications. It covers neural network and MCUs fundamentals, techniques for neural network optimization for resource-constrained environments, tools for end-to-end model deployment, and results on standard benchmarks.

**Audience:**

No

**Broader Impact Concerns:**

No concern on my end.

**Claims And Evidence:**

No

**Requested Changes:**

My suggestion to the authors is to cut section 2 to 4 entirely, and rewrite the review with focus on section 5 and 6 (deployment and on-device performance against standard benchmarks), while greatly expanding the comparison between competing MCUs, and between models on various benchmarks.

**Strengths And Weaknesses:**

STRENGTHS
- the topic of model deployment on extremely resource-constrained devices is very timely and of clear interest to the audience of this journal

WEAKNESSES
- in my opinion, this review totally misses the mark. It attempts to cover a wide range of topics, from neural network basics to on-device performance. In doing so, the treatment of each topics is shallow, incomplete, and ultimately of limited use for the reader.
- in terms of devices, this review discusses properties and on-device model accuracy of just 3 (three) MCUs from the ARM Cortex-M series. A fourth one is mentioned once (eDMPv1) and never brought up again.
- some sections appear disjointed from the topic of the paper: several topics are mentioned and described to various extent, but it remains unclear whether they found practical application in real-world scenarios. For example, the section on efficient models introduces pruning, distillation, quantization, etc. as viable (or necessary) strategies to compress the models within target constraints but when later discussing on-device performance, it is never mentioned whether any of these technique is used, nor what the trade-offs are.

OTHER WEAKNESSES:
- the content of some section is inconsistent: for example, Section 2.3 "Modern deep learning architectures" begins by listing common neural network building blocks, but it ends discussing regularization strategies. Section 2.2 "Properties" only discusses expressiveness and generalization. This section also mentions "double-descent" as a crucial phenomenon to be considered in relation to generalization, but no connection is made with TinyML applications and the topic is not mentioned again in the rest of the paper. Section 6 is titled "Limitation of TinyML" but the content is about model accuracy vs. model size.
- use of language is not incorrect (aside for the occasional typo) but the text could definitely benefit from extensive polishing

---

### Note · Authors · 2024-04-10

**Comment:**

Dear Reviewers and Editorial Team,
We would like to extend our sincere thanks for the constructive feedback provided on our manuscript. After careful consideration, we have concluded that the scope of revisions required to address your comments is substantial and would necessitate more time than is available within the two-week timeframe.
Given these circumstances, we have made the difficult decision to withdraw our paper from consideration. This will allow us to thoroughly revise our work, ensure it meets the high standards required for publication, and submit it elsewhere.
We appreciate the opportunity to have our work evaluated, thank you once again for your valuable insights.
Best,
the authors

**Withdrawal Confirmation:**

I have read and agree with the venue's withdrawal policy on behalf of myself and my co-authors.